# Operational and Analytical Modal Analysis of a Bridge Using Low-Cost Wireless Arduino-Based Accelerometers

**DOI:** 10.3390/s22249808

**Published:** 2022-12-14

**Authors:** Seyedmilad Komarizadehasl, Pierre Huguenet, Fidel Lozano, Jose Antonio Lozano-Galant, Jose Turmo

**Affiliations:** 1Department of Civil and Environment Engineering, Universitat Politècnica de Catalunya, BarcelonaTech. C/Jordi Girona 1-3, 08034 Barcelona, Spain; 2Acoustic, Vibration and Fluid Dynamics Discipline of SENER Company, Parc de l’Alba C/Creu Casas i Sicart, 86-87, Cerdanyola del Vallès, 08290 Barcelona, Spain; 3Department of Civil Engineering, Universidad de Castilla-La Mancha., Av. Camilo Jose Cela s/n, 13071 Ciudad Real, Spain

**Keywords:** Operational Modal Analysis, Arduino, low-cost accelerometer, IoT-based sensor, long-term monitoring, FDD, SSI-cov

## Abstract

Arduino-based accelerometers are receiving wide attention from researchers to make long-term Structural Health Monitoring (SHM) feasible for structures with a low SHM budget. The current low-cost solutions found in the literature share some of the following drawbacks: (1) high noise density, (2) lack of wireless synchronization, (3) lack of automatic data acquisition and data management, and (4) lack of dedicated field tests aiming to compare mode shapes from Operational Modal Analysis (OMA) with those of a digital model. To solve these problems, a recently built short-span footbridge in Barcelona is instrumented using four Low-cost Adaptable Reliable Accelerometers (LARA). In this study, the automatization of the data acquisition and management of these low-cost solutions is studied for the first time in the literature. In addition, a digital model of the bridge under study is generated in SAP2000 using the available drawings and reported characteristics of its materials. The OMA of the bridge is calculated using Frequency Domain Decomposition (FDD) and Covariance Stochastic Subspace Identification (SSI-cov) methods. Using the Modal Assurance Criterion (MAC), the mode shapes of OMA are compared with those of the digital model. Finally, the acquired eigenfrequencies of the bridge obtained with a high-precision commercial sensor (HI-INC) showed a good agreement with those obtained with LARA.

## 1. Introduction

The evaluation of bridge infrastructures in Italy after the Polcevera Viaduct collapse in Genoa served as a warning that sturdy construction alone cannot ensure a bridge’s longevity [1]. Morgese et al. [2] indicate that real-time health monitoring of this bridge could have provided sufficient information for adequate maintenance and warning of the impending failure 2–4 years before the actual bridge collapse in August of 2018. In fact, continuous monitoring of bridges is necessary to spot defects and damage and to schedule maintenance periodically [1]. It should also be noted that the health state of bridges normally depends on the country. For example, the USA’s infrastructure is strongly dependent on the state.

In reality, the pathologies (such as those associated with degradation of structural and mechanical properties of the materials) in the structures can advance substantially faster due to a lack of continuous maintenance and repair efforts or due to ineffective ones. It is essential to highlight that these pathologies can endanger the very stability of the structures [3].

The possibility of lowering the risks connected with structural pathologies depends on the feasibility of accurately analyzing its structural performance [4]. In addition, continuous maintenance and repair activities are required to ensure functional and safe conditions throughout the lifecycle of civil infrastructures and buildings [5]. One of the great challenges of long-term monitoring is distinguishing the modal parameter changes caused by environmental parameters (such as temperature and humidity) from those caused by structural pathologies. It is shown in the civil engineering literature (see, e.g., [6]) that the natural frequency of a structure decreases with the increase in ambient temperature and humidity. However, in comparison with the eigenfrequencies, these environmental parameters do not play an important role in mode shapes. In fact, to take into account the ambient variations, it is suggested by many scholars (such as in [7]) that SHM applications should be performed based on long-term average values. Among the applications for long-term ambient vibration assessment, it is important to highlight the Probabilistic Power Spectral Density (PPSD), which has been successfully used in several regions (such as India, Bulgaria, Italy, and Austria) [8].

Assessment of the health state of infrastructures throughout their lifecycle is essential to verify their structural safety and serviceability [9] and minimize their future reparation costs [10]. Structural Health Monitoring (SHM) applications can provide essential information about the current structural response, performance, and condition of infrastructures [11]. Consequently, SHM techniques are gaining extensive attention from civil engineers and infrastructure owners [12].

The Structural System Identification method is a vital component of SHM that evaluates the parameters of the structural model [13]. Set up from the structural response that was induced, Structural System Identification applications can be grouped as static [14,15] or dynamic [3,16,17]. Dynamic Structural System Identification techniques require the natural structure’s dynamic features (such as frequencies, damping ratio, and mode shapes) [18,19] that can be calculated by the modal analysis of the structure [20].

Modal analysis is typically used to determine infrastructures’ mechanical characteristics [21,22]. In the literature, three main modal analysis methods can be found. Experimental Modal Analysis (EMA): EMA is known to be the oldest modal analysis method and is carried out by artificially exciting a structure using an impact hammer or a shaker in a controlled environment such as a laboratory [23]. Operational Modal Analysis (OMA): this method performs a modal analysis of a structure under operation, which is only excited by its ambient vibrations [24]. Impact Synchronous Modal Analysis (ISMA): this application carries out the modal analysis of a structure under operation that is excited both artificially and by its ambient vibrations [25]. Unlike EMA, OMA and ISMA are performed while the structure is in its ordinary operation [26]. The difference between OMA and ISMA is that OMA is performed under ambient excitation without any additional artificial excitation instruments. The value of these vibrations, which will be the inputs of OMA, is unknown. Contrary to OMA, ISMA uses artificial excitation as in EMA, with known values as its input value [27].

It should be noted that OMA or EMA methods depend on the availability of the external excitation of a structure and are not dependent on the type of the sensors used for measuring the corresponding response of the structure. In fact, both contact and contactless sensors can be used for OMA and EMA. The selection of the sensor type (contact or noncontact) depends to a great extent on the structure characteristics, as well as on the monitoring budget.

Table 1 summarizes the comparison of the aforementioned modal analysis applications.

One of the well-known drawbacks of OMA is the inaccurate identification of harmonics as poles, which has been discussed in several works [28]. In addition, when the frequency of the input’s harmonic component is near to an eigenfrequency of the system, OMA approaches fail to appropriately identify the modal parameters [29].

The most significant advantage of OMA in comparison with EMA and ISMA is its modal characterization in natural operating conditions when it is difficult to artificially excite the structure [30]. Consequently, in civil engineering applications, OMA methods are typically used for bridges [31].

Some of the most common OMA methods in the literature are as follows: (1) Frequency Domain Decomposition (FDD), and (2) Stochastic Subspace Identification (SSI).

On the one hand, FDD is a nonparametric frequency domain modal analysis method that analyzes the structures under ambient excitations [32] and is widely used due to its simplicity [33]. This method is also known as Complex Mode Indicator Function (CMIF) to show its ability to detect multiple roots. This ability counts the number of primary (dominant) modes at a particular frequency. In addition, since the mode shape outputs of the FDD method are usually complex eigenvalues, they must be converted to real mode shapes [33]. After their conversion, the mode shapes of the operational analysis can be compared with an analytic model.

Stochastic Subspace Identification (SSI) is a parametric time domain modal analysis method that fits a parametric model directly into the raw timeseries data. In this approach, a parametric model is considered as a mathematical model that has parameters that can be changed to alter how the model fits to the data [34]. Two of the most popular methodologies of SSI are data-driven (SSI-data) and covariance (SSI-cov) [35]. The literature pointed out that SSI-data methods are less straightforward and computationally more expensive than the SSI-cov methods [36,37].

The basic idea behind frequency domain techniques is to calculate the modal parameters from quantities (such as magnitude, phase, and half-bandwidth) related to the spectral response function’s peak values. SSI algorithms, on the other hand, approach this problem from a time domain point of view, modeling the obtained timeseries as the output of an analogous linear system whose governing equations are entirely characterized by the linked state–space matrices [36]. The primary benefit of SSI approaches over traditional spectral alternatives is that they are entirely unsupervised [37].

To compare the mode shapes generated from a modal analysis technique with another technique, Modal Assurance Criterion (MAC) is typically used [38]. One of its uses is comparing an operational analysis’ mode shapes with an analytical one [39].

The needed vibration response of the structure for modal analysis is usually acquired using accelerometers [40,41]. These types of devices are force-based sensors that are typically used to measure the vibration response of structures. The four most common types of accelerometers are piezoelectric [42], piezoresistive [43], differential capacitive [44], and micro-electro-mechanical systems (MEMS) accelerometers [31]. Piezoelectric accelerometers with a range frequency of up to 12 kHz are among the most popular types of accelerometers traditionally used in SHM applications [45]. On the other hand, MEMS accelerometers are silicon-based sensors that found their way into many industrial applications due to their significant ongoing technological developments.

According to the literature [46], the cost of accelerometers is one of their main limitations for SHM applications. In fact, the price of the vibration acquisition equipment is not limited to the accelerometer itself since additional costs (data acquisition equipment, high-quality wires for transferring analog data, and a real-time controller) must be considered [31].

Furthermore, it should be noted that the focus of SHM applications is shifting from temporary structural assessment (also known as time-based evaluations) to long-term monitoring (permanent-based) applications. This shifting requires for low-cost, reliable instrumentation of the infrastructures [47].

It should be noted that an accelerometer’s sensitivity has an indirect relation with the squared value of the resonant frequency. As a result, the higher the sampling frequency of an accelerometer, the lower its sensitivity [48]. Consequently, to choose the right accelerometer, the dynamic characteristics of the structures under study should first be studied. It should be noted that the primary natural frequency range of most civil structures that are typically used for SHM is between 0.2 and 100.0 Hz [49]. It is shown in the literature on civil engineering that the primary natural frequency of short-span bridges (span length of a maximum of 40 m [50]) usually ranges between 3.0 and 30.0 Hz [51,52,53]. The primary eigenfrequency of the medium-span and long-span bridges is between 0.1 and 8.0 Hz (see, e.g., [54,55,56]). It should be noted that an accelerometer’s sensitivity has an indirect relation with the squared value of the resonant frequency. As a result, the higher the sampling frequency of an accelerometer, the lower its sensitivity [48].

In fact, MEMS accelerometers can offer a low-cost alternative with low-power consumption, high sensitivity, and an adequate sampling frequency compared with piezoelectric accelerometers [47]. Another advantage of using MEMS accelerometers is their connectivity to low-cost microcontrollers such as Arduino. Arduino, a low-cost microcontroller with several communication ports [57] (such as the Inter Integrated Circuit (I2C) serial communication bus), can be used to connect with a MEMS accelerometer [58].

It should be noted that more and more prototypes based on the use of low-cost MEMS accelerometers are being developed every day. For example, Pardeshi et al. [59] used an ADXL335 accelerometer and an Arduino Mega2560 to estimate the tire pressure of a vehicle using the vibration of the wheel hub. Patange et al. [60] developed a system for detecting various failure modes of a single-point cutting tool using machine learning and an Arduino-based vibration acquisition equipment that uses an ADXL335 MEMS accelerometer.

Notwithstanding the versatility and the advantages of MEMS accelerometers, they are typically affected by intrinsic noise density values comparatively higher than those from other types of accelerometers [37]. In addition, while piezoelectric accelerometers use expensive external precise signal conditioning circuitry [31], MEMS accelerometers typically incorporate it onboard. However, even though this onboard incorporation provides possible connectivity with low-cost microcontrollers (Arduino), it is a known source of additional noises [61].

To enhance the noise density of low-cost MEMS accelerometers, Komarizadehasl et al. [31] proposed combining several synchronized and aligned accelerometers and presenting the average value of the acquired vibrations as an output. This sensor combination reduced the inherent noise density of the individual sensors used and enhanced the noise density and resolution of the combined system.

The Low-cost Adaptable Reliable Accelerometer (LARA) is a triaxial accelerometer with improved noise density and accuracy by combining the outputs of five synchronized MPU9250 circuits using a multiplexor (TCA9548A). According to [31,47], reporting the averaged results of five similar low-cost accelerometers aligned and located on a rigid plate improves RMS resolution, noise density, and accuracy.

LARA is based on Arduino and Raspberry Pi technologies. Furthermore, LARA uses an activated Network Time Protocol (NTP) that can be post-synchronized for modal analysis applications. In addition, LARA has an open-source programable data acquisition application written in Python language [47].

A literature review comparing LARA with other developed low-cost MEMS accelerometers for SHM applications is summarized in Table 2. This table includes the following information collected in columns: (1) the name or the model of the accelerometer, (2) the acceleration range, (3) the sampling frequency, (4) the noise density (ND), (5) application, and (6) references.

Analysis of Table 2 shows that LARA has a better noise density (51 µg/√Hz) than most of the other developed solutions (usually ranging between 300 and 400 µg/√Hz). As a result, LARA distinguishes itself from most current low-cost accelerometers in the market. In fact, its accuracy and resolution can be compared to those of commercial accelerometers (PCB 393A03 and 356B18) [31].

It should be noted that contact sensors (such as accelerometers) are not the only type of sensor that can be used for the modal analysis of a structure. In fact, the literature review shows the increasing use of noncontact modal analysis of structures. Marwits et al. [67] investigated the feasibility of using three arrays of Scanning Laser Doppler Vibrometers (SLDV) for a noncontact OMA. Wang et al. [68] developed a novel vision-based modal testing application using three synchronized cameras. The application was then used for the OMA of a suspension footbridge. Another noncontact sensor used for modal analysis is 3D Laser Vibrometry. However, due to its high cost, it is normally used in EMA in the aerospace industry and when other solutions are not applicable, as in very light structures [69].

There is a gap in the literature regarding implementing low-cost Arduino-based accelerometers on actual structures under operation for mode shape assessment. This paper, for the first time in the literature, instruments a short-span bridge in Barcelona using four upgraded LARA accelerometers with automatized data acquisition features for OMA. After post-synchronizing the vibration acquisition of the used LARAs, a FDD and SSI-cov analysis is performed. Finally, the acquired complex mode shapes are normalized and compared with those of an analytical model of the bridge under study using the MAC equation.

In this experiment, the analyzed eigenfrequencies of the OMA are compared with those of a commercial dynamic sensor (HI-INC) [70].

The novelty of this paper refers to the use of a low-cost accelerometer prototype in an experiment with real conditions and comparing the mode shapes of the modal analysis with those of the digital model of the structure based on the available blueprints of the structure. This work also presents a novel application for LARA that automates its data acquisition and management.

It should be noted that the physical construction of LARA is part of the open-design movement that can provide new opportunities for other researchers to produce their own sensors instead of purchasing expensive commercial solutions. In order to encourage researchers to use LARA, the instructions for its manual alignment of 5 MPU9250 are presented in [47,71,72]. LARA is an open hardware prototype that has customization advantages. For example, in case a project needs temperature and humidity measurements, another sensor (such as DHT22 [73]) can be added to the system, making it versatile and customizable.

This paper is organized as follows: In Section 2, LARA is presented with its detailed characteristics and new upgrade for automated data acquisition and management. Further in this section, a laboratory experiment is performed to show the post-synchronization capability of LARA. Then, in Section 3, the analytical and operational modal analyses (FDD and SSI-cov) of a short-span bridge are presented with a detailed comparison of both methods. Finally, the main conclusions are drawn in Section 4.

## 2. Low-Cost Adaptable Reliable Accelerometer (LARA)

This section begins by explaining why LARA was chosen for this research. Then, a brief overview of the sensing and data acquisition components of LARA is provided. Finally, the post-synchronization application of this paper is presented using a laboratory experiment.

### 2.1. Selection of LARA

LARA is a low-cost wireless accelerometer with post-synchronization capability and a noise density of 0.00005 m/s^2^. This device has been validated in laboratory experiments against commercial experiments using commercial piezoelectric accelerometers (PCB 393A03 and PCB 356B18) [31]. In these laboratory experiments, the eigenfrequency measurement of LARA was validated in the frequency range of 0.4 to 32 Hz. Furthermore, its acceleration amplitude was confirmed within the range of 0.006 to 1 g (9.80655 m/s^2^). Moreover, the eigenfrequency analysis of LARA was validated on a short-span and a medium-span footbridge with a frequency range of 2 to 25 Hz [47].

Table 3 compares the RMS resolution and price of a LARA with some commercial accelerometers commonly used in SHM applications for bridges and footbridges. This table includes the following information collected in columns: (1) name, (2) price: based on a recent quote from retailors (VAT excluded), (3) sampling frequency, (4) broadband resolution, also known as Root Mean Square (RMS) resolution, (5) sensitivity, and (6) references.

It should be noted that the only wireless solution presented in Table 3 is LARA. Furthermore, unlike the commercial solutions, LARA does not necessitate the purchase of additional Data Acquisition Equipment (DAQ). This equipment adds an average extra cost of EUR 700 for every single measurement channel [31].

In addition, LARA in its current form has a higher broadband resolution than traditional commercial accelerometers (Table 3). Consequently, its use can be limited to those structures that have a high ambient vibration, such as the OMA of bridges, as they are normally dependent on the heavy ambient vibrations to excite the structure. It should be noted that to deal with unwanted frequencies such as ambient vibrations, bandpass filters from the signal processing toolbox [76] of MATLAB can be used. It should also be noted that the current noise density of LARA based on the presented information in Table 2 is 51 µg/√Hz. This value can be improved by aligning more MPU9250 chipsets on a single PCB. This is because the noise density of LARA has an indirect ratio with the square root of the number of aligned chipsets [31].

LARA has an adequate sampling frequency of 333 Hz for vibration acquisition of bridges, three times higher than the indicated eigenfrequency range of short-span bridges in the literature [49].

Figure 1a presents the LARA sensor used in this study. This device is composed of two essential parts: (1) the sensing part: containing the aligned accelerometers and the multiplexor, and (2) the data acquisition part: containing the Arduino used to convert the accelerometer outputs to a typical acceleration value (g) and the Raspberry Pi 3+ used to save the Arduino outputs and provide internet access to the entire system. It is critical to state that LARA employs Arduino and Raspberry Pi for robust data acquisition and post-synchronization of multiple accelerometers. The Arduino serves as a data conditioner, and the Raspberry Pi serves as a low-cost data acquisition system capable of providing wireless access to the data acquisition process and the low-cost accelerometer configuration with microsecond resolution timestamps [47].

Figure 1b shows the compact version of the LARA circuit developed by the research group with a dimension of 40 × 40 × 2 mm. It should be noted that LARA can be assembled by hand using commercially available MPU9250 circuits and a TCA9548A multiplexor. This manual assembly is fully detailed in [31].

It should be noted that the proposed methodology is not limited to Arduino-based sensors. In fact, the adaptation of the sensing part (to other IoT-based sensor solutions such as ESP32- and ESP8266-based microcontrollers) is a straightforward task.

### 2.2. Automatic Tailored Data Acquisition Application of LARA

This paper presents an upgraded version of LARA, automatizing its data acquisition process. Figure 2 shows the programmed steps of this, within the following four stages: (1) Initiation: after connecting the Raspberry Pi to any power source, it intends to connect to the Internet. (2) Cloud storage: Automatically mounts a Google Drive and downloads the latest Python code for the data acquisition process if an Internet connection is available. In the absence of the Internet, the present Python code on the hard drive of the Raspberry Pi will be used. (3) Vibration acquisition: at the scheduled date and time, the data acquisition begins, and (4) data management: the finalized acquired data are moved to the Google Drive when an Internet connection is available.

After connecting LARA to a power source, the Raspberry Pi is programmed to connect to the Internet with a Wi-Fi, LAN, or SIM Card. The next step is automatically mounting cloud storage (Google Drive) on its hard drive. This way, each LARA will have access to a specific folder in the drive where it can download and save Python code for its data acquisition procedure. Later, LARA starts the vibration acquisition at the scheduled date and time indicated on the received Python code

Furthermore, the acquired vibrations are saved in a text file on the Raspberry Pi hard drive for 30 min. It should be noted that this duration can be altered inside of the data acquisition Python code. Another task is written on the operating system of Raspberry Pi to periodically check and move the text files containing the data acquisition to the Google Drive. The hard drive is never overburdened or filled as long as the Internet connection (through Wi-Fi or LAN) is available. It is essential to mention that in case of the absence of the Internet connection, the acquired data will stay on the hard drive of the Raspberry Pi until the Internet connection is fixed or the memory card is full. It should be noted that the storage can be upgraded using external hard drives.

It should also be noted that Raspberry Pi and its programmed tasks can face errors or malfunctions can happen. These unforeseen software problems are fixed by rebooting the system. The system reboot can fix issues such as connection problems with Arduino, absence of a response from the accelerometer, and lack of access to the Google Drive. For that, LARA is programmed to be rebooted once every day. During the short-term data acquisition applications, the flow of acquired data must be checked periodically, and in case of any issue, a reboot must be manually applied.

### 2.3. Post-Synchronization Application of LARA

To show the post-synchronization capability of LARA, a laboratory experiment was designed. In this experiment, two LARAs were located on the midspan of a simply supported beam model. Then, several impacts were introduced on the midspan of the beam model. The overlapping of the acquired vibration of the LARAs used can show the synchronization of both sensors.

To perform this experiment, a U-shaped aluminum profile with a section dimension of 25 × 25 × 3 × 3 mm and an adequate length of 1100 mm was chosen. Figure 3a shows the setup of LARA 1 and LARA 2 on this beam. The access to the data acquisition of these LARAs was provided from a Wi-Fi 4G SIM Card router. As presented in Figure 3a, LARA 1, LARA 2, and the used router were directly connected to separate power banks. After mounting the sensors on the beam using an industrial adhesive, the sensors were scheduled to start a vibration acquisition simultaneously.

Figure 3b illustrates on the same graph the data acquisition outputs of LARA 1 and LARA 2. The analysis of this figure shows that even though the sensors started their vibration acquisition simultaneously, a lag appeared 150 s after the data acquisition initiation. The reason behind this lag is the production imperfection of the data conditioner used (Arduino Due). Even though the provider of the purchased Arduinos, the length and material of the used wires, and the uploaded code to the bios of both Arduinos are the same, the sampling frequency of LARA 1 (333.33 Hz) was found to be different from the sampling frequency of LARA 2 (333.85 Hz). This difference resulted in about 0.2 s of lag drift between the two devices.

To fix this issue, a MATLAB code was used to resample the acquired data of the accelerometers. This code is part of the signal processing toolbox of MATLAB and is written to change the sampling frequency of uniform or nonuniform data to a new one [76].

Figure 3c presents the resampled acquired vibrations by LARA 1 and LARA 2 in a time domain representation figure. The sampling frequency of both LARAs is now the fixed value of 333 Hz.

It is essential to mention that if the sampling frequency of two accelerometers is not exactly the same, mode shapes obtained by carrying out a modal analysis cannot be trusted. To solve these issues, resampling functions are typically used.

A resampling function can modify the sample rate of time domain data. This is a typically used method in the vibration analysis literature [77], for frequency conversion and resampling of timeseries data. There are various methods for resampling time domain data [76]. In this paper, a MATLAB function that uses a polyphase antialiasing filter for resampling the time domain data at a uniform sample rate (333 Hz in this paper) was used. This function is shown in Equation (1):(1)$$Y=resample\;\left(x,t_{x},\;Fs\right)$$

In Equation (1), *Y* refers to the time domain series with a steadily fixed sampling frequency of 333 Hz, x presents the time domain series with a slightly different sampling frequency (such as LARA 1 with a sampling frequency of 333.33 Hz), $t_{x}$ is the corresponding timestamps of x, and Fs is the desired final sampling frequency, which here was set to be 333 Hz.

Analysis of Figure 3 shows the importance of the resampling program for synchronizing the acquired vibrations of several low-cost accelerometers based on the Arduino technology. It should be noted that without the activated NTP of the Raspberry Pis and their timestamps with microsecond resolution, the resampling could not have been accurately performed. It should be noted that the current experiment was only carried out to show the apparent lag in the time domain representation. This is because the sampling frequency of each LARA was slightly (about 0.1%) different than the other. Notwithstanding that the sampling frequency of each LARA is calculated before performing an eigenfrequency analysis, the frequency domain representation of each LARA is accurate. However, the application for performing a modal analysis using a number of synchronized LARAs requires a unique sampling frequency. This means that all the accelerometers used must have the same sampling frequency. To fix the individual sampling frequency of each LARA, their acquired vibrations were resampled with a sampling frequency of 333 Hz.

Nonetheless, the eigenfrequency analysis of LARA has previously been validated through 11 experiments on an actuator in the structural laboratory of the Universitat Politècnica de Catalunya (UPC) university, Barcelona, Spain [47,71]. In that work, it was reported that eigenfrequency analysis of LARA had 0.5% and 0.003% for signals with 0.1 and 32 Hz, respectively.

It should also be noted that the results of the eigenfrequency analyses of LARA 1 and LARA 2 before and after fixing their sampling frequency should be the same. To demonstrate this, eigenfrequency analyses were performed on the time domain responses presented in Figure 3b,c and Figure 4a,b, which show the eigenfrequency analysis of LARA 1 and LARA 2, respectively, before fixing their sampling frequency using the resampling MATLAB function. Furthermore, Figure 4c,d show the eigenfrequency analyses of LARA 1 and LARA 2, respectively, after fixing their sampling frequency at the rate of 333 Hz.

The analysis of Figure 4 shows that fixing the sampling frequency of the sensors does not affect the eigenfrequency analysis results. In fact, the highest difference between the detected frequencies in Figure 4a,b is less than 0.04%. It can also be seen that Figure 4c,d represent the same frequency value.

It should be noted that the sampling frequency of the accelerometer used should have a relation with the frequency of the understudy signal. This relation (also known as Nyquist frequency) indicates that a repetitive waveform can be accurately reconstructed if the sampling frequency is higher than twice the maximum frequency that has to be sampled [35]. Moreover, knowing the exact sampling frequency of the accelerometer used is another important topic that is needed for performing an eigenfrequency analysis. Additionally, it should be kept in mind that, when performing a modal analysis using a number of synchronized accelerometers, the sampling frequency of the accelerometers used should be the same. This is because the programs aiming to perform a modal analysis normally ask for a single sampling frequency and not the exact sampling frequency of each accelerometer used.

## 3. Experimental Modal Analysis of a Short-Span Bridge

This section compares an experimental modal analysis of a footbridge in Barcelona using four LARAs with an analytical modal analysis. The findings of this test were validated by comparing the eigenfrequency analysis of LARA with those of a commercial dynamic sensor. In addition, the generated mode shapes of the operational modal analysis were compared with an analytical FEM model (SAP2000 model [62]). Firstly, the preparation of the digital model of the structure is presented. Then, modal analysis of the structure using the outputs of the LARA accelerometers is detailed. Finally, the results of the experimental and analytical models were compared with each other.

### 3.1. Bridge Description and Structural Modeling

This section provides the characteristics of the studied footbridge. Then, the bridge’s structural modeling is illustrated. Further in this section, the description of the different elements that make up the theoretical model of the pedestrian walkway is presented. The primary aspects considered when modeling the bridge are then outlined. Following that, the sensor positioning is indicated. Finally, the findings of the bridge model’s modal analysis are discussed.

First, the studied structure (Polvorines’ Footbridge in Barcelona, Spain) is shown in Figure 5a. As shown in this figure, the structure is connected to an elevator box and on an abutment. Figure 4c and Figure 5b present the section and plan of the bridge, respectively. This bride is 13.68 m long and 2.35 m wide. The primary beams are IPE600 beams located in the footbridge’s longitudinal direction. In addition, the transverse beams are IPE160. The deck of this bridge is composite and has been modeled in the software with the shell-thin element with a thickness of 120 mm.

This structure was selected because of its relatively reduced uncertainty (the structure is made mainly of steel, it was built recently (2019), and all its blueprints were available in [78]).

From Figure 6a, it is clear that the bridge is connected to the elevator box using a shear tap [79]. Figure 6b shows the other support of this footbridge. As shown in Figure 6b, the bridge can freely move along its longitudinal axis. In fact, only the translation *Z* and *Y* directions are restrained on these supports.

A numerical model of this structure was generated in the software SAP2000 using the available drawings of the bridge. The physical and mechanical properties of the material used in the FEM were defined based on the construction blueprints of the walkway. Table 4 shows the characteristics of the materials used.

In Figure 7a, the location of the accelerometers (4 LARAs) is shown. This way, after the modal analysis of the software, mode shapes can be compared with the analytical research. The location of the commercial dynamic sensor (HI-INC beanair) is shown in Figure 7a. HI-INC is a dynamic inclinometer that has a sampling frequency of 250 Hz [47]. This sampling frequency enables HI-INC to acquire the main mode shapes of the footbridge under study. Figure 7b shows the setup of LARA 1 and HI-INC on the midspan. It should be noted that LARA accelerometers are mounted to this structure with their X and Y axes parallel to the X and Y axes (Figure 7a) of the footbridge. Consequently, the *Z*-axis of the accelerometers is aligned with the vertical axis of the bridge and the gravitational force of the earth. The sensors LARA 1, Hi-INC, and LARA 4 were located in the midspan (6.84 m) of the structure support and the sensors LARA 2 and LARA 3 were located in the one-fourth (3.42 m) of the bridge abutment.

Table 5 shows the frequency, mode shapes, and mass participation output of the SAP modal analysis for the first four mode shapes. Ux, Uy, and Uz indicate the mass participation on longitudinal, transversal, and vertical directions of the bridge, respectively. Rx, Ry, and Rz show the rotational mass participation of the bridge’s longitudinal, transversal, and vertical axes, respectively.

The analysis of Table 5 shows that the primary mode shapes with the highest mass participation are modes 1, 2, and 3. In addition, their eigenfrequency is lower than 1/10 of the sampling frequency of LARA. Furthermore, it can be seen that these three mode shapes are only vertically active.

Table 6 presents the node displacements corresponding to the location of the mounted accelerometers for the first three vertical mode numbers together with the normalized mode shapes of the digital model under the modal analysis load case. Table 6 also presents the mode shape of every mode number.

The analysis of Table 6 shows that this bridge is designed to work like a simply supported beam. In fact, the first and the third mode shapes are bending mode shapes. In addition, no torsional mode shapes are obtained. Further analysis of the information of this table shows that the eigenfrequencies of this bridge are higher than 5 Hz. This makes this bridge safe from the resonance effects that can happen because of the walking frequency of a person (usually between 2 and 3 Hz) and the first harmonic (approximately 5 Hz) which is caused by the double excitation caused by the interaction of the heel and toe [80].

### 3.2. Operational Modal Analysis of the Bridge under Study

In this section, the modal analysis of the bridge under study is presented.

For the present work, two alternative OMA methods were applied to analyze the measurement data obtained by the LARA sensors. After mounting the LARAs on the bridge under study (Figure 7) using an industrial adhesive [31], each was connected to a power bank. Right after being connected to the power bank, the sensors were booted automatically, and then connected to a previously introduced wireless Internet connection. They started their data acquisition process as scheduled and their data acquisition files were constantly uploaded to the Google Drive. After two hours, the sensors were turned off as they were coded and detached from the bridge.

After successfully post-synchronizing and resampling the data of all LARAs, the acceleration vectors were used in an open-access MATLAB code for the FDD analysis and the post-normalization and conversion of the generated complex mode shapes [81]. Then, the synchronized acceleration vectors were used in another open-access MATLAB code for the SSI-cov analysis [82].

#### 3.2.1. OMA Using FDD

Since the analytical model’s first three generated mode shapes only affect the *Z*-axis (vertical axis), only the *Z*-axis data of all accelerometers have been analyzed in detail. The outputs of the FDD analysis are summarized in Figure 8 for *Z*-directions.

Figure 8 shows that the first three mode shapes are very evident and selecting them is performed by choosing the highest peaks of the figure.

Table 7 shows modal values of the first three generated mode shapes of the experimental analysis. The information in this table includes: (1) LARA number: corresponding to a LARA located on the bridge, (2) mode shape: presenting the mode number of the modal analysis, (3) frequency: reporting the eigenfrequency of FDD for every step number, (4) complex mode shapes: showing the raw mode shapes of the FDD analysis for every accelerometer, and (5) normalized real mode shapes: after converting and normalizing the generated complex mode shapes of FDD using Reference [83], this information can be compared with the normalized mode shapes of the analytical model.

#### 3.2.2. OMA Using FDD

Since this bridge’s first three mode shapes were found in the Z-direction by the modal analysis of the digital model, the SSI-cov was performed on the same axis. The SSI-cov method was used to calculate the stabilization diagram of the identified eigenfrequencies as a function of the model order. The stabilization diagram is presented in terms of the number of poles in Figure 9.

The analysis of Figure 9 shows that the highest picks are selected by the program itself. Unlike FDD, SSI methods select the picks themselves. The studies carried out showed that the SSI-cov required a lower computational capacity in comparison with the FDD one. Table 8 presents the calculated eigenfrequencies and normalized mode shapes obtained by the SSI-cov method.

### 3.3. Comparison of the Results

To compare the outputs of the structural model with the results of the OMA methods, the eigenfrequencies and mode shapes of the first three vertical mode numbers were compared with each other. This comparison also includes an eigenfrequency analysis carried out using a commercial dynamic sensor. The mode shapes of the analytical analysis with the experimental study have been compared using the Modal Assurance Criterion (MAC) value [39]. The MAC value is calculated using Equation (2):(2)$$MAC\left(r,q\right)=\frac{|{\left\{\varphi_{r}\right\}}^{T}{\left\{\varphi_{q}\right\}|}^{2}}{\left({\left\{\varphi_{r}\right\}}^{T}\left\{\varphi_{r}\right\}\right)\left({\left\{\varphi_{q}\right\}}^{T}\left\{\varphi_{q}\right\}\right)}$$

In Equation (2), $\varphi_{r}$ refers to the normalized mode shape matrix extracted from the FEM model and $\varphi_{q}$ is the extracted mode shape matrix from FDD analysis, which is already converted to real mode shape values and is normalized. The value of the MAC number must be between 0.8 and 1 for every mode shape to show a remarkable resemblance of the mode shapes [39].

Table 9 shows the error of the acquired frequency for every mode number from the commercial dynamic sensor used (HI-INC). Table 9 also shows the MAC number of the first three mode shapes of the bridge under study. The MAC value comparison of the mode shapes of FDD analysis with those of the analytical analysis and mode shapes of SSI-cov analysis with those of the analytical model are shown in columns MAC_FDD, Digital_ and MAC_SSI-cov, Digital_, respectively.

Comparing the measured eigenfrequencies of LARA with those of HI-INC from Table 9 showed a maximum difference of 0.87 Hz for the third mode, representing a difference of 3.3%. However, compared with the outputs of other developed low-cost accelerometers in the literature (accelerometers in Table 1), 3.3% looks like a great benchmark.

In addition, the analysis of Table 9 showed that the eigenfrequency analysis of FDD and SSI-cov methods presented similar outputs. In fact, the maximum difference between the illustrated eigenfrequencies was less than 0.05 Hz for the first mode, representing a difference of 0.6%.

Further analysis of Table 9 showed that the first reported eigenfrequency of the SAP2000 model was 16.4% off from the reported value of the high-precision instrument (HI-INC). The difference between the first analyzed eigenvalue of the digital model from that of HI-INC might be due to the use of a shear tab for connecting the bridge beams to the elevator box. In fact, a number of civil engineering scholars report that the shear tab causes rotational rigidity [79].

Table 9 also showed that all the presented MAC values were within the range of 0.93 to 1.00. This shows a very good correlation between the analytical model and the OMA methods. Consequently, it can be deduced that LARA sensors are sensitive and accurate enough for OMA applications and can be used in FDD and SSI-cov methods.

The analysis of Table 9 showed the good correlation of the FE model with the bridge. This is because the mode shapes extracted from the modal analysis of the SAP model, correspond accurately to those mode shapes extracted from the OMA of the footbridge.

## 4. Conclusions

Developing and validating low-cost sensors is critical for Structural Health Monitoring (SHM) of structures with limited funds for SHM assessments. In addition, with low-cost sensors, the long-term monitoring of infrastructures can be feasible and affordable. Previous works introduced a Low-cost Adaptable Reliable Accelerometer (LARA), where its eigenfrequency analysis was validated in the laboratory and on two footbridges (one flexible and one less flexible). This system is a low-cost triaxial accelerometer with a sampling frequency of 333 Hz with attached data acquisition equipment.

This paper first presented a program to automatically perform the data acquisition and management of LARA. This program was made to be self-sufficient. In fact, as long as the Internet connection (Wi-Fi or LAN) is available and the power source provides the sensors with electricity, this system acquires the vibration of the bridge under study based on its programmed schedule and moves the received data to a Google Drive folder. Furthermore, this program was written to perform self-troubleshooting. Every day at a known time, the device gets rebooted, receives the newest Python program for the data acquisition process (if available), and starts the automatic vibration acquisition process.

To validate the performance of the developed sensor in real conditions, an operational modal analysis on a footbridge in Barcelona was carried out using four LARA accelerometers. After post-synchronizing the acquired vibrations from the four LARAs, they were used in a parametric and a nonparametric OMA. The parametric modal analysis used was the Covariance Stochastic Subspace Identification (SSI-cov) method. This method was chosen because of its cheap computational requirement and its high accuracy. In addition, the modal analysis of the bridge using the Frequency Domain Decomposition (FDD) method, which is a nonparametric modal analysis application, was performed.

Finally, the first three vertical mode shapes and eigenfrequencies of this footbridge’s OMAs (SSI-cov and FDD) using four LARAs were compared with a numerical model. The mode shapes generated from the digital model have been compared with the mode shapes of the modal analysis using the Modal Assurance Criterion (MAC). The MAC values comparing the modal analysis of the digital model and SSI-cov were between 0.93 and 1.00. In addition, the comparison of mode shapes generated from the modal analysis of the analytical model with FDD showed MAC values ranging between 0.95 and 1.00. These MAC values for the first three mode shapes illustrate LARA’s high accuracy and reliability in experimental modal analysis. Furthermore, the generated frequencies of the OMA from the four LARA accelerometers were validated with a commercial dynamic sensor (HI-INC).

The main limitations of LARA are as follows: (1) Noise density: LARA has a higher noise value (51 µg/√Hz) in comparison with traditional wireless commercial accelerometers (usually ranging between 8 and 30 µg/√Hz). Even though the current resolution and accuracy of LARA are enough for the SHM application of bridges, they are not enough for the analysis of buildings. To improve the noise density of LARA, more MPU9250 chipsets can be aligned. (2) Data curing: The number of needed post-processing procedures for preparing the acquired vibrations for performing modal analysis. In future works, those post-processing codes aiming to synchronize and fix the sampling frequency of individual accelerometers should become automated.

To summarize, both the laboratory and in situ tests carried out in this work validate the applicability of LARA for the SHM of the studied bridge.

## Figures and Tables

**Figure 1 sensors-22-09808-f001:**
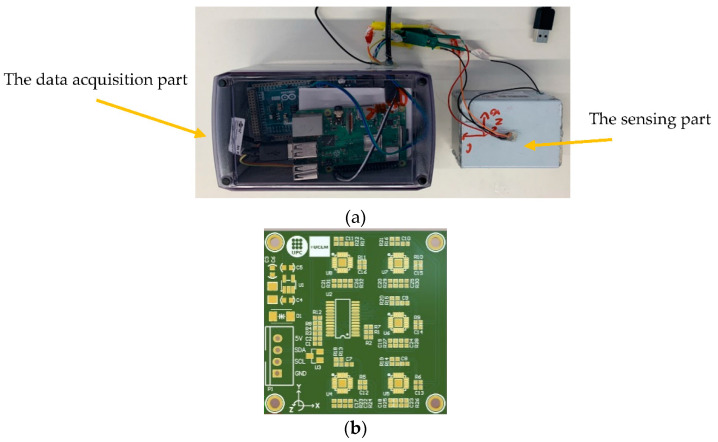
LARA: (**a**) boxed sensing part and data acquisition part of LARA, and (**b**) PCB design of LARA.

**Figure 2 sensors-22-09808-f002:**
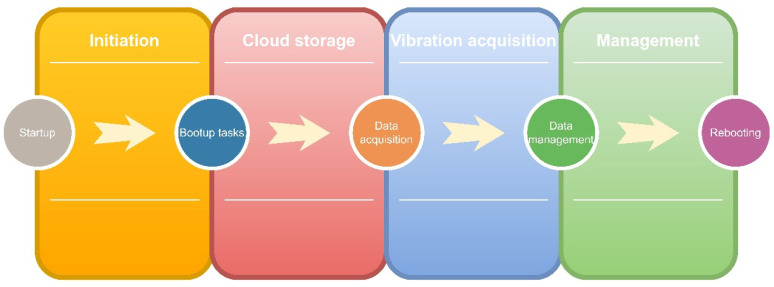
Automation of the data acquisition process of LARA.

**Figure 3 sensors-22-09808-f003:**
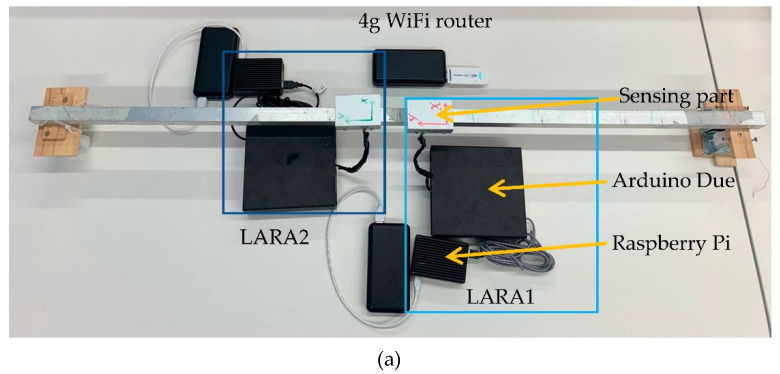
Synchronization experiment: (**a**) setup of two LARAs on a simply supported beam, (**b**) time domain vibration presentation of two LARAs with the same initiation time, and (**c**) time domain vibration presentation of two LARAs with the same initiation time and resampled acquired vibrations.

**Figure 4 sensors-22-09808-f004:**
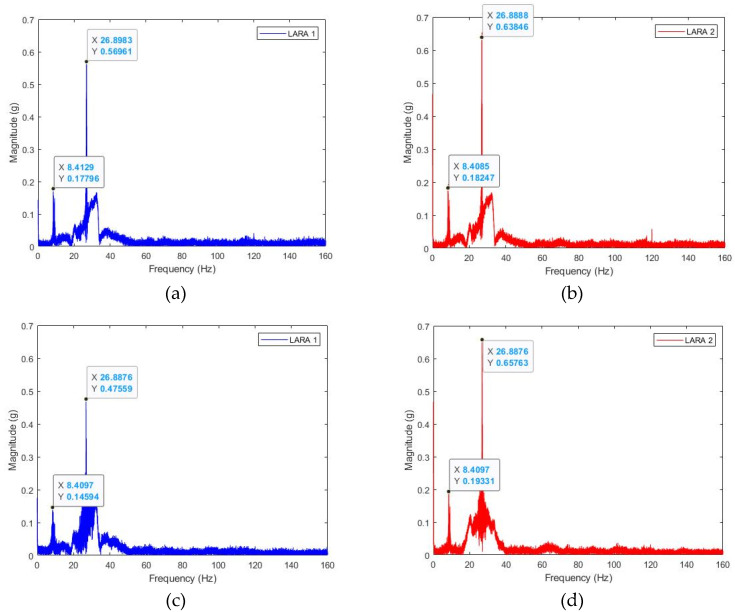
Eigenfrequency analysis of LARA 1 and LARA 2 located on a simply supported beam model: (**a**) LARA 1 before being resampled, (**b**) LARA 2 before being resampled, (**c**) LARA 1 after being resampled, and (**d**) LARA 2 after being resampled.

**Figure 5 sensors-22-09808-f005:**
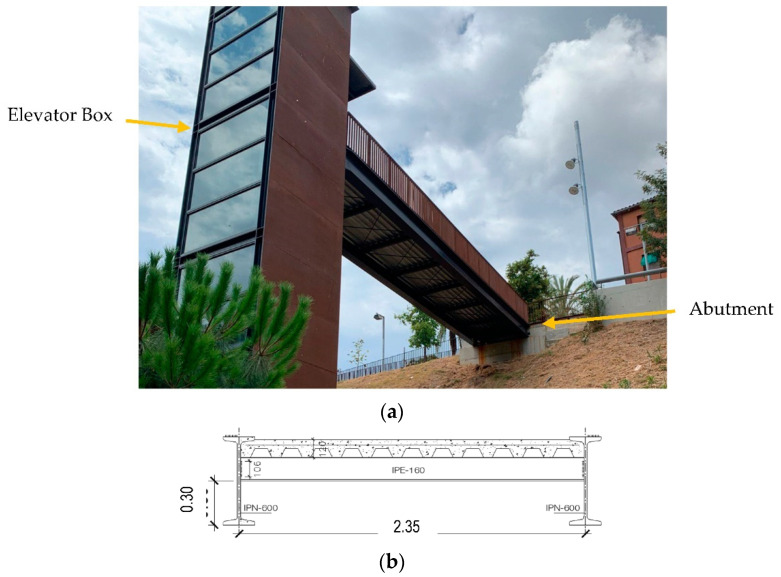
Polvorines footbridge: (**a**) a photo of the bridge, (**b**) the bridge’s cross-section, and (**c**) the bridge’s plan (all units are in meters).

**Figure 6 sensors-22-09808-f006:**
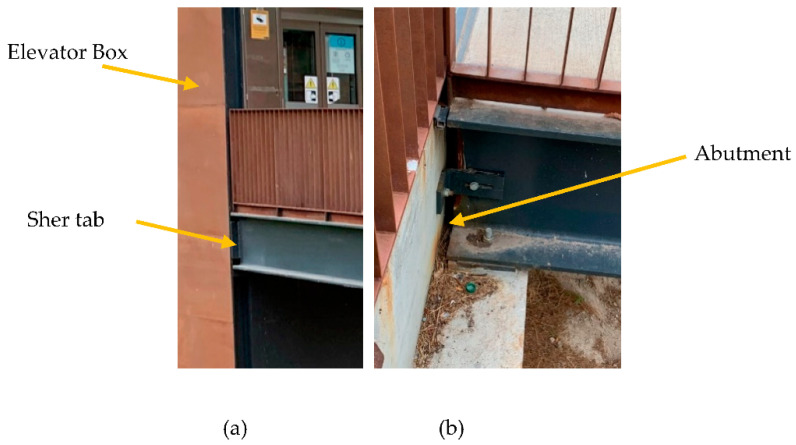
Supports of the footbridge bridge: (**a**) connection to the elevator and (**b**) abutment connection.

**Figure 7 sensors-22-09808-f007:**
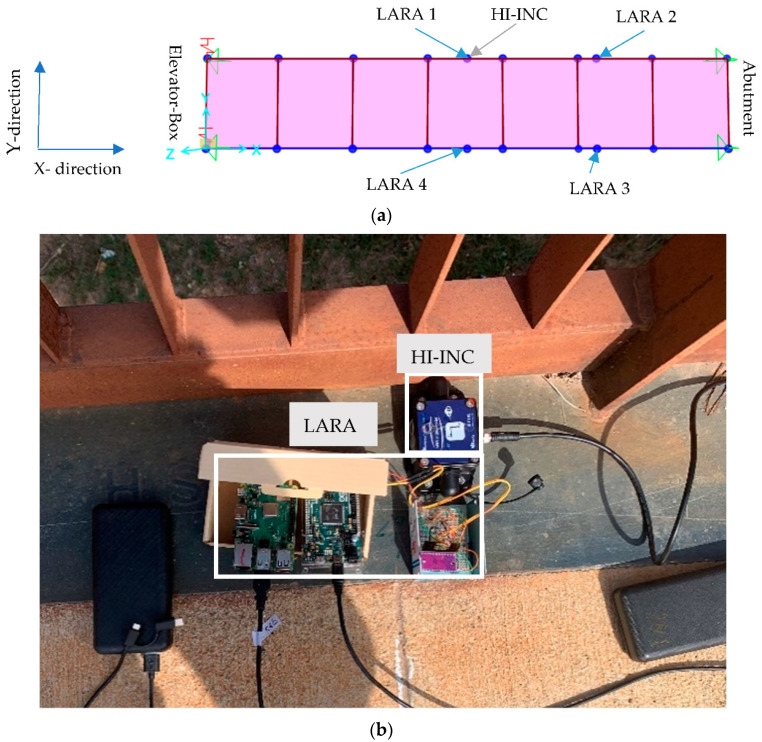
Digital model and position of the sensors: (**a**) SAP2000 model of the footbridge under study and allocation of the used sensors, and (**b**) instrumentation of the Polvorines footbridge.

**Figure 8 sensors-22-09808-f008:**
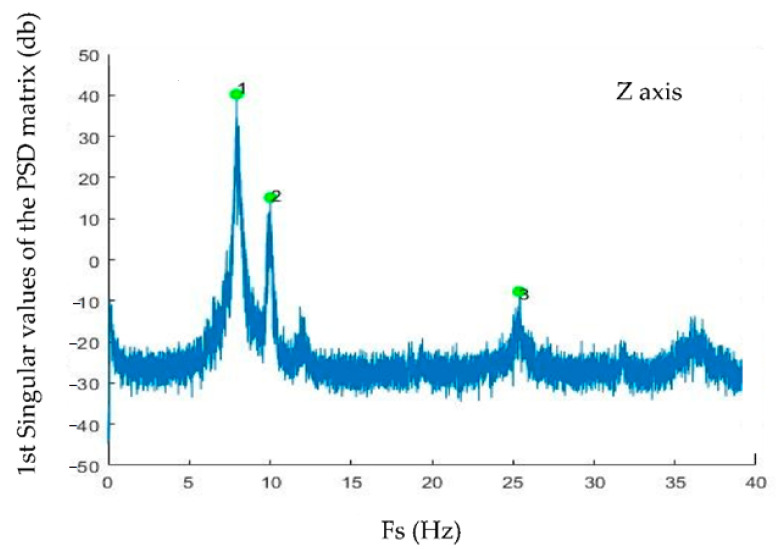
Modal analysis of a footbridge by FDD on the *Z*-axis.

**Figure 9 sensors-22-09808-f009:**
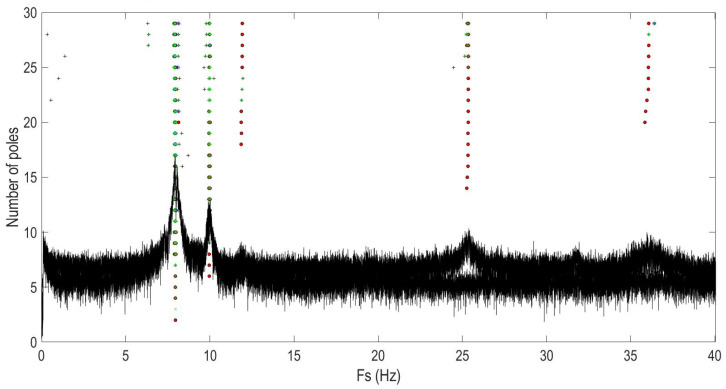
Stabilization diagram of the identified eigenfrequencies calculated with the SSI-cov.

**Table 1 sensors-22-09808-t001:** Comparison of the famous modal analysis methods.

Modal Analysis Method	Natural Excitation of the Structure under Ambient Vibrations	Artificial Excitation of the Structure by Using a Vibrator or an Impact Hammer
EMA		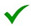
OMA	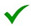	
ISMA	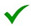	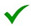

**Table 2 sensors-22-09808-t002:** Low-cost solutions in the literature to measure accelerations.

Accelerometer	Acceleration Range (g)	Sampling Frequency (Hz)	Spectral Noise (µg/√Hz)	Application	References
ADXL335	±3 g	100	300 ^1^	Eigenfrequency analysis of a bridge	[59,60]
LIS344ALH	±2 g	100	50 ^1^	Mode shape assessment of a beam model	[62]
LEWIS	±2 g	100	400 ^1^	Shaking table that simulates transverse displacements	[63]
LEWIS2	±2 g	500	300 ^1^	Shaking table that simulates transverse displacements	[64]
LSM9DS1	±2 g	952	No data	Eigenfrequency analysis of a nearby sheet piling work	[65]
ADXL362	±2 g	100	400 ^1^	Mode shape assessment of a laboratory floor	[66]
LARA	±2 g	333	51 ^2^	Eigenfrequency analysis of a bridge	[47]

^1^ Datasheet. ^2^ Laboratory test.

**Table 3 sensors-22-09808-t003:** Comparing a LARA with standard accelerometers for the SHM of bridges.

Name	Price	Sampling Frequency	Broadband Resolution	Sensitivity	Type	Reference
	EUR	Hz	mg RMS	V/g		[47]
LARA	140 ^3^	333	0.93	0.625	Triaxial	
IMI 604B31	613	5000	0.35	0.100	Triaxial	[74]
IMI 607A61	324	10,000	0.35	0.100	Uniaxial	[75]

^3^ Research prototype.

**Table 4 sensors-22-09808-t004:** Material characteristics of the footbridge.

Material	Description	Characteristics	
1	S275 beam steel		
		Elasticity modulus (MPa)	210,000.0
		Poisson’s ratio (ν)	0.3
		Specific weight, ρ (kN/m^3^)	78.5
2	Concrete		
		Elasticity modulus (MPa)	24,855.0
		Poisson’s ratio (ν)	0.2
		Specific weight, ρ (kN/m^3^)	25.0

**Table 5 sensors-22-09808-t005:** Modal mass participation of the bridge under study.

Mode Number	Fs Hz	Ux Unitless	Uy Unitless	Uz Unitless	Rx Unitless	Ry Unitless	Rz Unitless
1	6.61	0	0	0.895	0	0.008	0
2	9.83	0	0	0	0.895	0	0
3	24.56	0	0	0.003	0	0.825	0
4	30.25	0	0	0	0.003	0	0

**Table 6 sensors-22-09808-t006:** Normalized mode shapes of SAP2000 from the modal analysis load case.

LARANumber	Mode Number	Frequency	Normalized Amplitude	Mode Shape
LARA 1	1	6.61 Hz	1.00	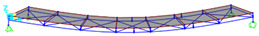
LARA 2	0.71
LARA 3	0.71
LARA 4	1.00
LARA 1	2	9.83 Hz	−0.99	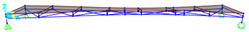
LARA 2	−0.85
LARA 3	0.85
LARA 4	1.00
LARA 1	3	24.56 Hz	0.05	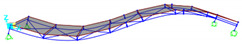
LARA 2	1.00
LARA 3	1.00
LARA 4	0.05

**Table 7 sensors-22-09808-t007:** Modal analysis output of the FDD application for the four low-cost sensors.

LARA Number	Mode Number	Frequency	Complex Mode Shapes	Normalized Real Mode Shapes
LARA 1	1	7.924 Hz	−0.581 + 0.000i	−1.000
LARA 2			−0.343 + 0.229i	−0.710
LARA 3			−0.145 − 0.332i	−0.624
LARA 4			−0.580 − 0.025i	−0.999
LARA 1	2	10.023 Hz	−0.542 − 0.000i	−0.857
LARA 2			−0.232 + 0.263i	−0.555
LARA 3			0.284 + 0.029i	0.451
LARA 4			0.632 − 0.009i	1.000
LARA 1	3	25.383 Hz	−0.014 − 0.000i	−0.022
LARA 2			0.455 − 0.048i	0.732
LARA 3			0.268 − 0.566i	1.000
LARA 4			−0.001 + 0.060i	−0.097

**Table 8 sensors-22-09808-t008:** Modal analysis output of the SSI-cov application for the four low-cost sensors.

LARA Number	Mode Number	Frequency	Normalized Shapes
LARA 1	1	7.966 Hz	−0.999
LARA 2			−0.582
LARA 3			−0.691
LARA 4			−0.999
LARA 1	2	10.007 Hz	0.663
LARA 2			0.999
LARA 3			−0.851
LARA 4			−0.628
LARA 1	3	25.341 Hz	−0.048
LARA 2			0.615
LARA 3			1.000
LARA 4			−0.060

**Table 9 sensors-22-09808-t009:** Eigenfrequency and MAC value comparison of the experimental analysis with the digital model.

Mode Number	Digital Model (Hz)	HI-INC (Hz)	FDD (Hz)	SSI-cov (Hz)	MAC_FDD, Digital_	MAC_SSI-cov, Digital_
1	6.61	7.91	7.92	7.97	1.00	1.00
2	9.83	9.94	10.02	10.01	0.95	0.93
3	24.56	26.25	25.38	25.34	0.96	0.93

## Data Availability

Not applicable.

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
