# Peer review of "Operational and Analytical Modal Analysis of a Bridge Using Low-Cost Wireless Arduino-Based Accelerometers"

_sensors, 2022, doi:10.3390/s22249808_

Round 1

Reviewer 1 Report

Dear authors,
In the beginning, I would like to congratulate you on a potentially very interesting paper with a comprehensive approach to developing low-cost sensors. Below I would like to point out some elements worth improving.
A)    General remarks
1.    The article is not always clearly written and easy to follow.
2.    The authors give relevant references which are linked to their study.
3.     The abstract is well written introducing the basic overview of the paper. It is also written in a way that even a person not familiar with the topic can understand what the authors are proposing in their research. However, it is advised not to present specific data in the abstract and focus more on presenting the novelty of the study.

4. The introduction requires some improvements. The purpose of the introduction is to present the problem of the article and clearly present the overview of the state of the art in case of the topic and presented later on methods. Some problems noticed: a. The first paragraph is regarding the poor state of bridges in the USA. This is one case but USA infrastructure is quite different than the European one and strongly depends on the state. Maybe present another source of data. Easy to find is an evaluation of bridge infrastructure in Italy after the Genova bridge collapse. It would be an improvement to add in this paragraph the information that some civil engineering (CE)structures are being constantly monitored and present some monitoring methods. E.g for long-time vibration monitoring of CE structures (https://doi.org/10.1016/j.engstruct.2005.09.001 or./and DOI:10.1007/s13349-020-00442-z ) and especially PPSD method for civil engineering structures monitoring (e.g DOI: 10.21008/j.0860-6897.2020.3.11 ) b. The part about EMA and OMA is good and it is correct that the following part is about measurement techniques. However, this part is incomplete. The authors present only contact methods. However, there is a whole other set of sensors like microphone arrays or especially contactless optical methods for EMA and OMA using e.g. 3D Laser Vibrometry Systems. EMA Example: DOI: 10.1109/IDAACS53288.2021.9661060 and OMA DOI:10.1007/978-3-319-30084-9_7. Additionally, table 1 can be extended with a column about this method to the comparison look more professional. Since the authors are using later a contact method it is enough to add just a short paragraph about non-contact measurements.

5.    Chapter 2:
a.    Fig.3 – presenting in milli g not really professional, seconds unit small “s”, not capital “S”.
6.    Chapter 3
a.    Something is wrong with the editing and information between lines 384-393
b.    Fig 6b suggest enlarging
7.    Results are clearly presented however the noise lever is quite high and probably will not allow measuring above the first three natural frequencies. If so please indicate what are possibilities to limit the noise or present this information as a system limitation.
8.    Conclusions are acceptable. Please just perform editing according to the template. The font size is smaller than the rest of the text.

B)    Conclusions
The article is clear and interesting with no significant errors found in the research but with some questions that need to be answered. Both methodology and results acquisition is mostly correct. Some additional state-of-the-art analysis of modern modal analysis testing techniques has to be incorporated in the introduction. The article has potential after eventual improvements. At the current stage, the reviewer asks for minor changes in those areas and will be happy to accept the paper after those corrections.

Author Response

The authors sincerely appreciate the positive comments of the two Reviewers and are very grateful for their suggestions and observations. These comments certainly have improved the quality of the paper. In addition, they gave us valuable hints on our future research. Detailed responses to each of the reviewer's comments are provided in the attached document. The answers of the authors are highlighted in green, the new information is in blue and the excluded information in red.

Reviewer 2 Report

Why was a frequency-domain response not considered for data represented in Figure 3? I can see only time-domain response.

How was the sampling frequency set? Justify.

More discussion must be provided on the effect of sampling frequency on data acquisition.

How to deal with heavy noise during data acquisition? Comment on the robustness of this system proposed.

Justify accelerometers arrangement, orientation, and placement.

It is good to add specific inferences drawn from modal analysis of the structure to justify its design.

What’s the limitation of this work? Please add it.

The scope of a MEMS-based accelerometer integrated with an open source controller, such as Arduino, etc. for measurement of vibrations should be extended by referring to the articles ‘Application of Machine Learning for Tool Condition Monitoring in Turning,’ and ‘Tyre Pressure Supervision of Two Wheeler Using Machine Learning'. You can add this in the literature review or Table 2 (ADXL335).

In continuation, try connecting your work to the open-design movement reviewed in the paper. This would open new opportunities for authors and readers as well.

The article is lovely but needs some revision, as suggested herein, for final recommendation. All the best!

Author Response

(The authors gave the same response as above.)

Round 2

Reviewer 2 Report

Congratulations. The authors have addressed all my comments positively.